# Deep Learning in Breast Cancer Imaging: State of the Art and Recent Advancements in Early 2024

**DOI:** 10.3390/diagnostics14080848

**Published:** 2024-04-19

**Authors:** Alessandro Carriero, Léon Groenhoff, Elizaveta Vologina, Paola Basile, Marco Albera

**Affiliations:** Radiology Department, Maggiore della Carità Hospital, 28100 Novara, Italy; alessandro.carriero@maggioreosp.novara.it (A.C.); vologina.elizaveta@gmail.com (E.V.); pa.basile.25@gmail.com (P.B.); marco.albera@icloud.com (M.A.)

**Keywords:** artificial intelligence, deep learning, breast cancer imaging

## Abstract

The rapid advancement of artificial intelligence (AI) has significantly impacted various aspects of healthcare, particularly in the medical imaging field. This review focuses on recent developments in the application of deep learning (DL) techniques to breast cancer imaging. DL models, a subset of AI algorithms inspired by human brain architecture, have demonstrated remarkable success in analyzing complex medical images, enhancing diagnostic precision, and streamlining workflows. DL models have been applied to breast cancer diagnosis via mammography, ultrasonography, and magnetic resonance imaging. Furthermore, DL-based radiomic approaches may play a role in breast cancer risk assessment, prognosis prediction, and therapeutic response monitoring. Nevertheless, several challenges have limited the widespread adoption of AI techniques in clinical practice, emphasizing the importance of rigorous validation, interpretability, and technical considerations when implementing DL solutions. By examining fundamental concepts in DL techniques applied to medical imaging and synthesizing the latest advancements and trends, this narrative review aims to provide valuable and up-to-date insights for radiologists seeking to harness the power of AI in breast cancer care.

## 1. Introduction

Breast cancer is the most common neoplastic disease in women, with over 2.3 million diagnoses and 685,000 deaths registered globally in 2020 [1].

Screening programs play a pivotal role in recognizing breast cancer in preclinical stages, allowing less invasive but radical treatment and therefore improving outcomes in terms of overall survival and quality of life. Breast cancer screening programs rely on radiological examinations, mainly mammography (MG) and ultrasonography (US), for detecting early signs of neoplasm [2], such as microcalcifications, architectural distortions, and solid masses. While these programs have greatly improved the detection and prognosis of breast cancer, the ever-increasing workload and possibility of false positives and negatives have prompted research for supporting tools able to improve diagnostic performance. The challenge of this critical task lies in its severely attention- and time-consuming nature, which is key to avoiding missing even the finest details in large amounts of high-resolution images analyzed throughout each workday. Excellent focus and consistent performance are paramount skills for medical imaging specialists, but even the best human operators will ultimately be limited by factors such as fatigue, biases, and distractions. In this context, appropriately trained AI algorithms can either be used as second or third independent readers to provide failsafe mechanisms or as real-time assistants to enhance radiologists’ sensitivity and specificity, representing the ultimate advancement in computer-aided detection (CADe). Potentially, they could even be implemented as automated single readers to increase reporting speed, reduce costs, and therefore enlarge screening audiences, but this approach carries complex bioethical implications still to be addressed by regulators.

After the tumor has been detected, staging and cancer burden monitoring with magnetic resonance imaging (MRI) might be required in selected cases [3]. Traditional staging approaches include bi-dimensional lesion measurements and infiltration assessment, which often rely on subjective judgment and therefore induce significant interobserver variability [4]. AI-based tools provide automated or semi-automated lesion identification, potentially delivering more consistent and reproducible results, and therefore allowing for more precise staging [5]. Moreover, they can significantly reduce the time required for performing such measurements, making it much easier and faster to compare different studies and evaluate treatment response or disease progression [6].

Finally, radiomics-based models have been used to predict key clinical information, such as histopathological features, prognosis, and treatment response, from medical imaging examinations [7], by analyzing quantitative image patterns hidden from human qualitative observation. Despite still being mostly confined to research applications, these computer-aided diagnosis (CADx) approaches may radically accelerate and enhance the implementation of personalized medicine in the near future.

CAD instruments based on image feature analysis and statistical classifiers have been available in medical imaging software suites for several years, but issues such as low specificity severely hindered their adoption. Automated image interpretation research, also known as computer vision (CV), has recently been revolutionized by deep learning, a new subset of artificial intelligence and machine learning techniques based on multi-layer neural networks mimicking the human brain architecture. Demonstrating much higher performance than previous solutions, convolutional neural networks (CNNs) have become the most established DL-based approach for complex CV tasks and have been extensively adopted in the medical imaging realm.

Several DL-based models for breast cancer imaging, both commercial and open source, have thus become available in recent years and have been favorably tested in comparative studies. However, their clinical usage is not yet widespread due to existing limitations and challenges [8], such as reproducibility of results, costs, explainability, privacy, and liability.

## 2. Contribution, Novelty, and Motivation Statement

In this narrative review, we present fundamental concepts about deep learning in medical and breast cancer imaging and describe the latest advancements in this field. Specifically, we will examine key studies published in recent years and delve into novel developments regarding both technical elements and study design. Finally, we will summarize current results, limitations, and challenges of AI-assisted breast cancer imaging.

We acknowledge that extensive research literature has already been produced on this topic, both in terms of experimental studies and review articles. However, we have found two partly unexplored themes that we would like to highlight and further develop.

Firstly, we observed that most existing reviews have focused on either clinical [9], or technical aspects [10]. In our work, primarily geared towards radiologists, we strive to achieve a middle ground by giving essential technical information regarding AI development, along with up-to-date research and clinical results. We believe this approach will stimulate medical imaging professionals’ interest in AI, helping them to better understand AI-related literature and more effectively leverage final software applications.

Secondly, we noted that previous similar works mostly assessed deep learning applications for conventional imaging modalities, such as MG, US, and MRI, while few have mentioned novel techniques, such as thermography and microwave-based imaging. We aim to provide a comprehensive overview of the role of DL in both clinically consolidated and investigational breast imaging examinations, to stimulate clinical interest as well as technological progress towards new diagnostic approaches.

Moreover, we firmly believe in the added value of providing continuous updates on this quickly evolving field. Recently published prospective studies conducted on large cohorts have shed new light on the potential and criticalities of deep learning applied to breast cancer screening, while technical advancements are constantly increasing deep learning performance and potential use cases in the biomedical realm. By briefly examining these novelties, we hope to provide an up-to-date overview of the topic while outlining existing limitations and challenges.

## 3. Deep Learning in Medical Imaging: Approaches and Techniques

DL is a subset of AI techniques that employs trained neural networks for completing various tasks [11]. Unlike older machine learning approaches, which could only evaluate a hard-coded collection of features in a certain object, deep learning neural networks automatically select the most relevant features to extract and combine based on the training data. This provides a more versatile and general-purpose applicability and simplifies the creation of new models [12].

As highlighted in Figure 1, the most common applications of medical imaging deep learning models involve processing images to either infer an output result, such as the presence of abnormal findings, or manipulate their characteristics, e.g., reducing noise and improving spatial resolution. Other ancillary uses include report generation and information research. We will briefly examine common DL techniques used in medical imaging and then explore in more detail current DL applications for breast cancer imaging.

A detailed discussion about neural network development is outside the scope of this review and is largely researched by non-medical specialties. However, we believe that a basic understanding of the main technical characteristics of deep learning models would be beneficial to all medical imaging professionals, to better assess their performance and limitations and help their adoption in clinical practice.

### 3.1. Convolutional Neural Networks

As illustrated in Figure 2, CNNs are feedforward neural networks that consist of multiple convolutional layers followed by pooling layers and fully connected layers [13], designed for processing grid-like structures such as images thanks to an architecture inspired by the animal visual cortex. Convolutional layers apply filters or kernels to extract features from local regions of input images. These filters can detect edges, shapes, textures, and patterns, which serve as building blocks for object recognition. After several convolution operations, the output feature maps undergo downsampling through max-pooling, average-pooling, or min-pooling layers to reduce spatial resolution while retaining salient information. Finally, fully connected layers perform high-level reasoning to produce classification results. In the medical imaging field, CNNs are mainly used for image classification, object detection, and segmentation. Figure 3 provides a simplified representation of these tasks.

#### 3.1.1. Classification and Object Detection

Classification models evaluate an image categorizing it under a labeled class, such as normal or abnormal. Neural networks commonly used for image classification include Residual Networks (ResNet) [14], Densely Connected Convolutional Networks (DenseNet) [15], MobileNets [16], EfficientNet [17], and ConvNeXt [18]. Object detection models expand on the classification concept, not only recognizing whether an image is abnormal or not, but also identifying the approximate location of the abnormality, usually by outputting a rectangular bounding box. This information greatly increases the understandability of AI-proposed assessments for radiologists and clinicians. Neural networks commonly used for object detection include You-Only-Look-Once (YOLO) [19], region-based convolutional neural network (R-CNN) [20], and single-shot multibox detection (SSD) [21].

Classification and object detection have been effectively applied to several tasks in screening and emergency settings, where they could help to select high risk patients that warrant extended review by a human observer. They have particularly been used with bi-dimensional imaging modalities, such as X-ray scans, automating detection of abnormalities like bone fractures [22], pneumonia [23], pleural effusion [24], pneumothorax [25], and neoplasms [26].

In recent years, classification models have also been used to analyze biomedical images predicting histopathological features, prognosis, and treatment response [27]. We will discuss the role of deep learning-based radiomics in more detail in later sections.

#### 3.1.2. Segmentation

Segmentation models identify the exact boundary of an object in an image or volume [5], allowing precise calculation of physical properties such as diameters, surface area and volume, X-ray attenuation, or signal intensity. These models are extremely useful for lesion burden estimation, surgical planning, and image-guided therapy. They are particularly adept at volumetric imaging techniques such as CT and MRI, with whom precise manual identification of multiple, variably shaped lesions would be extremely time-consuming.

Semantic segmentation aims to categorize pixels according to the classes they represent without distinguishing instances of the same class [28]. In other words, all pixels belonging to the same class are assigned the same label regardless of whether they correspond to separate objects or overlapping regions, providing a global understanding of the image content and distinguishing foreground from background. Semantic segmentation is particularly useful in applications where identifying the presence and location of specific classes is sufficient, e.g., tumor detection in medical images [29].

Instance segmentation [30], on the other hand, seeks to identify and delineate individual objects of the same class separately, enabling item counting and individual properties measurement. Applications requiring assessment of individual entities, such as teeth evaluation in panoramic X-ray images [31], can employ instance segmentation to perform more advanced analyses on each detected object.

Radiomics relies on accurate image segmentation to define regions of interest (ROIs). ROI selection determines the subsequent extraction of quantifiable attributes representing tissue characteristics, such as shape, density, and texture [7]. Subsequently, modeling strategies, like machine learning algorithms, link derived radiomic signatures to phenotypic traits, therapeutic responses, or molecular profiles. Thus, precise and consistent segmentation constitutes a cornerstone in reliable radiomic studies. Enabling highly accurate and fast automatic segmentations, deep learning has the potential to greatly reduce the time required for retrieving radiomics data from imaging studies [27].

The most used neural network for biomedical image segmentation is U-Net [32], which offers excellent support for three-dimensional datasets. No-New-U-Net (nnU-Net) is a publicly available, fully automated pipeline for creating U-Net-based models which have been successfully used for various tasks [33], such as whole-body segmentation from CT scans [34], brain cancer segmentation from MRI scans [35], kidney tumor segmentation from contrast-enhanced CT scans [36], and many more.

### 3.2. Generative Adversarial Networks

Generative adversarial networks (GANs) comprise a generator, which synthesizes artificial samples mimicking the distribution of real data, and a discriminator, which distinguishes between genuine and fake instances. During training, both components engage in a minimax game where they alternatively optimize their objective functions until equilibrium is reached. Once converged, the generator becomes proficient at generating realistic samples indistinguishable from actual ones, whereas the discriminator achieves near-perfect accuracy in discerning true versus false examples [37].

In the realm of medical imaging, GANs have mainly been used for synthetic data generation and image enhancement. GANs can generate plausible synthetic cases conditioned on known pathologies or risk factors, augmenting existing datasets and alleviating data scarcity issues [38]. Such augmented data can help train more accurate and robust predictive models, particularly when dealing with imbalanced classes or rare events. GANs can also refine noisy or suboptimal images, enhancing diagnostic visibility and reducing ambiguity [39]. For instance, GANs can accentuate subtle details, suppress artifacts, and harmonize inconsistent features across multi-center studies [40].

### 3.3. Large Language Models

Large language models (LLMs) have recently been explored for their potential applications in the field of medical imaging, specifically for tasks such as radiology report generation and information retrieval. These models are trained on vast amounts of textual data and can generate coherent and contextually relevant sentences [41], making them well suited for natural language processing (NLP) tasks in the medical domain.

Radiology reports are detailed documents that summarize the findings of medical imaging examinations. Producing these reports requires significant expertise and sometimes takes a lot of time. Large language models can automate this process by generating written descriptions of the observed abnormalities based either on textual input from a radiologist or directly on the analysis of medical images [42]. These generated reports can assist radiologists in interpreting complex studies and increase efficiency by reducing turnaround times [43].

Large language models are also able to provide quick access to relevant medical literature, guidelines, or clinical trial information based on specific inputs [44]. For example, when presented with certain imaging findings, a large language model could retrieve relevant articles, studies, or treatment options that may aid in diagnosis and management.

Several general-purpose LLMs have become widely used in recent years, both commercial, such as OpenAI’s GPT [45], and public, such as Meta’s LLaMA [46]. Basing on them, multiple domain-specific LLMs have been trained to better perform in the medical and radiology field, such as Med-PaLM [44], MedAlpaca [47], and LaVA-Med [48], some of them even integrating image assessment capabilities and therefore fully automating the reporting workflow.

### 3.4. Technical Considerations: Training, Inference and Deployment

As illustrated in Figure 4, the development of a deep learning model involves the selection of an appropriate neural network pipeline and the collection of an adequate training dataset. During the training phase, the neural network extracts and selects relevant features based on provided ground truth data, developing weighted connections between its neurons that eventually make it able to perform inference on new input data by itself. In recent years, with the availability of general-purpose, efficient neural networks and self-configuring pipelines [49], together with several medical imaging datasets [50], the creation of performant models has been significantly simplified.

Datasets containing a sufficiently large number of validated images manually labeled by radiologists as ground truth have become a precious asset for developers to train and test the performance of their models [50]. However, the usage of large public datasets poses challenges due to the heterogeneity of demographic characteristics and scanning devices, possibly hampering their performance in local facilities [51]. Conversely, a model trained to perform effectively on images acquired in a single institution for a specific patient demographic might not fare well when tested on a large public dataset or on other external datasets [52]. It is still debated which of the two solutions is the most suitable to improve results in final applications, but local retraining by transfer learning seems to be an effective approach [52]. Furthermore, persistent monitoring of AI efficacy through feedback collection and routine retraining might be required to guarantee adequate and sustained performance [53].

Advanced AI algorithms necessitate substantial computational capabilities, especially in terms of high-performance GPUs or other accelerators for training and inference [54]. Having these resources on-site allows for performing DL tasks directly in healthcare facilities. However, due to the high cost of such components, many commercial AI solutions offload computations to external servers, significantly reducing hardware expenses and local energy usage but introducing peculiar issues due to data transmission, namely privacy concerns, needing appropriate management [55].

To increase ease of use, AI-based insights should be seamlessly integrated into Picture Archiving and Communication Systems (PACS) [56]. Indicators such as malignity scores, heatmaps, and bounding boxes help radiologists understand inference results [57]. Commercial platforms for AI-based medical image interpretation provide graphical user interfaces requiring little to no technical knowledge to be operated, while open-source medical imaging models are often very difficult to use and implement in a clinical workflow. To date, no open-source, unified interface for medical imaging inference has been released. This is unlike other AI applications, where open platforms, such as llama.cpp for LLMs [58], and ComfyUI for GANs [59], allow end-users to run pre-trained models locally with low technical effort.

### 3.5. Performance Metrics for Medical Imaging Deep Learning Models

Accuracy measures the proportion of correct predictions out of all the predictions made. While it can be a useful measure in certain cases, it may not always provide an adequate representation of the model’s performance, especially if the dataset is imbalanced [60].

Precision measures the proportion of true positive predictions among all positive predictions. High precision indicates that the model has a low false positive rate, meaning that when it predicts a sample as positive, there is a high likelihood that it is indeed positive [60].

Sensitivity, also known as recall, measures the proportion of actual positive samples that were correctly identified by the model, calculated as the number of true positives divided by the sum of true positives and false negatives. A high recall value suggests that the model is good at identifying all relevant positive samples, even if it may generate more false positives [60].

The F1 score is the harmonic mean of precision and recall, providing a single measure that balances both these metrics [61]. Ranging from 0 to 1, with higher values indicating better performance, the F1 score is particularly useful when dealing with imbalanced datasets where one class is significantly underrepresented compared to the other [60]. In cancer-enriched medical imaging datasets, this can partly reduce the effect of underrepresenting negative cases.

The receiver operating characteristic curve (ROC) plots the true positive rate against the false positive rate for different classification thresholds [60]. The area under this curve (AUC) represents the probability that a randomly chosen positive sample will be ranked higher than a negative sample. An AUC = 1 implies perfect separation between positive and negative classes, while a value closer to 0.5 indicates poor discrimination ability. The AUC is often used to evaluate the performance of classification models [61].

The Dice Similarity Coefficient (DSC) is a measure of overlap between predicted segmentation masks and ground truth masks, commonly used for evaluating segmentation models [60]. It ranges from 0 to 1, with higher values indicating greater overlap and thus better segmentation performance [61].

## 4. Deep Learning in Breast Cancer Imaging: Datasets

As described in the technical discussion, high-quality and sufficiently large datasets have become crucial in the training and assessment of deep learning models for medical imaging. Due to the extremely significant healthcare burden posed by breast cancer and the large number of examinations performed for its screening, several datasets for breast cancer imaging have been publicly released. Table 1 lists the principal ones, most of them containing high-resolution screening mammograms [62], while only a few include US and MRI examinations.

The Digital Database for Screening Mammography (DDSM) was the first publicly available dataset developed to aid research in breast cancer computer-aided detection (CAD) using single-field mammography (SFM) images [63]. Released in 1999, it paved the way for pioneering works in analyzing and interpreting digital mammography scans. Comprising roughly 10,000 digitized film mammography studies, DDSM covers a broad spectrum of breast densities, ages, and health conditions. Each examination is categorized as either normal or containing benign calcification, mass, or microcalcification clusters. Associated reports detail diagnostic conclusions, radiologist interpretations, and follow-ups. Despite being considered somewhat dated, DDSM retains historical significance owing to its impactful contributions to the evolution of medical image analysis and CAD systems. Nevertheless, limitations arise from the digitalization process, introducing inconsistent resolution and signal-to-noise ratio issues significantly reduced in contemporary FFDM datasets.

INBreast was the first publicly available, high-quality digital mammography dataset, introduced in 2011 and consisting of 410 images from 115 subjects [64]. Compared to older datasets, it offered standardized, uniform, and high-resolution FFDM images, with detailed annotations of breast masses, microcalcifications, and benign/malignant categorizations. It also included metadata such as breast composition, age, and invasiveness indicators. Professional radiologists thoroughly examined each subject and annotated relevant ROIs accompanied by descriptive attributes, and two independent experts validated the annotations, reinforcing reliability. Importantly, INBreast reflects European populations, diversifying regional representativeness and supplementing established American datasets.

The Curated Breast Imaging Subset of DDSM (CBIS-DDSM) is a carefully selected subset derived from the DDSM [65]. It addresses shortcomings present in the original dataset, namely inconsistent compression rates, varying resolutions, and ambiguous ground truth annotations. The goal was to deliver a more coherent, standardized dataset compared to its predecessor. Key highlights of CBIS-DDSM include improved image quality through upscaling and sharpening techniques, uniform resolution and bit depth, and the absence of overlapping patches. Initially published in 2017, the CBIS-DDSM dataset comprises approximately 6700 studies drawn from the DDSM dataset, including 753 calcification cases and 891 mass cases. Every study comes with expert-reviewed annotations and accompanying metadata.

Introduced in 2022, VinDr-Mammo comprises 30,000 full-field digital mammography RoIs extracted from Vietnamese women, covering a wide array of breast densities, ages, and health conditions [66]. Each case was reviewed to identify suspicious findings, which were subsequently marked with bounding boxes. Metadata about age, breast density, acquisition view, and pathologic confirmation status, when available, was also included, offering potential training material for prediction models, radiomics, and decision-support tools. Not all included cases were confirmed with histopathological correlation; therefore, data interpretation mostly relied on the radiologists’ judgment.

The Annotated Digital Mammograms and Associated Non-Image (ADMANI) datasets are a curated collection containing more than 4.4 million screening mammography images from 630,000 women [67], including both image and associated nonimage data, confirmed by histopathological correlation. Introduced in late 2022, they constitute one of the largest mammography collections currently available. Notably, a subset of 40,000 images from 10,000 screening episodes has been donated as a test set for the RSNA Screening Mammography Breast Cancer Detection challenge [68].

Unlike screening mammography, only a few public datasets for ultrasound and magnetic resonance imaging examinations have been publicly released, and most of them are limited by either small sample size or suboptimal image quality. This has probably played a role in the slower development and assessment of deep learning models for these techniques. Recently released non-mammographic datasets include BrEaST [69] and BUS-BRA [70] for ultrasound and Duke Breast Cancer [71] and BreastDM [72] for dynamic contrast-enhanced magnetic resonance imaging (DCE-MRI).

Larger, private datasets, such as those from the New York University (NYU) for breast cancer screening [73], ultrasound [74], and magnetic resonance imaging [75], have also been commercially released, but the entry costs constitute a significant adoption barrier for worldwide researchers.

## 5. Deep Learning in Breast Cancer Imaging: Applications to Conventional Techniques

As illustrated in Figure 5, a rapidly increasing number of publications have explored the role of deep learning in breast cancer imaging. Key studies involving this field have been listed in Table 2 and will be explored in the next sections.

### 5.1. Conventional Mammography

Breast cancer detection with mammography has been one of the most prominent applications of deep learning techniques in the medical imaging field. As illustrated in Table 3, several commercial products have been made available and received regulatory certifications from the American Food and Drug Administration (FDA), becoming effectively approved for clinical use, and multiple retrospective and prospective studies have been conducted to assess their performance.

In a 2017 retrospective study by Becker et al. [76], a deep artificial neural network (dANN) from a commercial image analysis suite was used to aid in the detection of breast cancer in mammograms from a total of 1144 patients, 143 of which had histology-proven invasive breast cancer or another clinically significant lesion. The neural network was trained on a dataset of mammograms manually marked by radiologists, augmented with various transformations, and tested against an external dataset of patients with cancer and a matched control cohort, demonstrating comparable performance to experienced radiologists. In a screening-like cohort, the sensitivity/specificity of the dANN was 73.7/72.0%, with an overall diagnostic accuracy, expressed as AUC, of 0.82. The diagnostic accuracy was highest in low-density breasts, with an AUC = 0.94. However, the study also noted limitations of the dANN, including its lack of understanding of multiple views per patient and time evolution.

In a 2019 retrospective study by Watanabe et al. [77], a set of 2D full-field digital mammograms (FFDMs) was collected from a community healthcare facility in Southern California and interpreted using an AI-CAD software. The cancer-enriched dataset included examinations from 122 patients with 90 false-negative mammograms obtained up to 5.8 years prior to diagnosis and 32 BIRADS 1 and 2 patients with a 2-year follow-up of negative diagnosis. The study reported significant improvement in cancer detection rate (CDR) for all radiologists who used the software. The overall mean reader CDR increased from 51% without assistance to 62% with AI-CAD, with a less than 1% increase in the readers’ false-positive recalls. However, the study also found that the sensitivity of all readers appeared to be elevated due to the test setting and the enrichment of the dataset with a high proportion of abnormal mammograms.

Another 2019 retrospective study by Akselrod-Ballin et al. evaluated the performance of a combined machine learning–deep learning model for early breast cancer prediction using a large, linked dataset of electronic health records and digital mammography examinations from over 13,000 women [78]. The authors aimed to determine if the model could achieve a level comparable to radiologists and be accepted in clinical practice as a second reader. All available clinical features for each woman in the linked dataset, including characteristics previously recognized as risk factors for breast cancer, were extracted and evaluated by the algorithm. For the malignancy prediction task, the algorithm obtained an AUC = 0.91 with 77.3% specificity at 87% sensitivity, demonstrating an assessment level comparable to radiologists.

In 2020 Schaftter et al. published the results of a challenge that asked participants to develop algorithms outputting a malignity likelihood score based on screening mammography data [79]. Examinations from over 85,000 US women (952 cancer positive ≤12 months from screening) were used for training and internal validation, while a second independent cohort, including data from 68,000 Swedish women (780 cancer positive), was used for external validation. Thirty-one teams submitted their models for final validation, with the top performer achieving an AUC = 0.858 and 0.903 on the internal and external validation dataset, respectively. The standalone algorithm specificity measured at the radiologists’ sensitivity was 66.2% on the internal validation dataset and 81.2% on the external validation dataset, worse than both American (90.5%) and Swedish (98.5%) radiologists. However, combining top-performing algorithms and radiologists’ assessments resulted in a higher AUC (0.942) and achieved a significantly improved specificity (92.0%) at the same sensitivity. Overall, despite no single AI algorithm being able to outperform imaging specialists in this study, the collaboration between radiologists and an ensemble algorithm demonstrated the potential to decrease the recall rate by 1.5%.

A retrospective, multi-reader study published in 2020 by Kim et al. compared the diagnostic performance of humans, AI-aided humans, and standalone AI in detecting breast cancer on mammograms [80]. A commercial AI-based software was used to interpret 320 mammograms (160 cancer-positive, 64 benign, 96 normal). The performance level of AI measured as AUC was 0.940, significantly higher than radiologists without AI assistance (0.810). With the assistance of AI, radiologists’ AUC was improved to 0.881. Compared to radiologists, AI was more sensitive in detecting cancers with mass, distortion, or asymmetry and T1 or node-negative neoplasms. Overall, this study found that AI can potentially achieve similar or better performance to medical imaging experts. However, the authors reported several limitations, primarily the use of a cancer-enriched dataset, which has different cancer prevalence to real-world data.

In 2020, the results of another large retrospective study conducted on Swedish women were published by Dembrower et al. [81], aiming to examine the use of an AI cancer detector to triage mammograms and identify women at the highest risk of undetected cancer. The final study sample included 547 diagnosed patients and 6817 healthy controls from the Karolinska University Hospital uptake area. The AI-based software analyzed each screening mammogram and generated a numerical score related to the likelihood of cancer signs in the image. Women with a score below a rule-out threshold were triaged to the no radiologist work stream, while those with a score above a rule-in threshold (after negative double reading by radiologists) were triaged to an enhanced assessment work stream. According to the authors, many women could be appropriately triaged by the AI-based software alone without missing any cancer that would have been detected by radiologists. Furthermore, by assigning suspicious cases to additional imaging evaluations (e.g., MRI), AI pre-emptively recognized a significant proportion of subsequent interval cancers and next-round screen-detected cancers. These results suggested that AI could potentially halve human workload while also detecting a substantial amount of early human-missed neoplasms.

Between 2021 and 2022, Dembrower et al. conducted a prospective, population-based, non-inferiority study, whose results were published in 2023 [82]. The authors evaluated double reading by two radiologists against double reading by one radiologist plus AI, single reading by AI, and triple reading by two radiologists plus AI. The study population consisted of more than 55,000 women examined at Capio Sankt Göran Hospital in Stockholm, Sweden. Women with breast implants, a known genetic mutation, a very high lifetime risk, or a personal history of breast cancer were excluded. Double reading with AI plus one radiologist proved to be non-inferior to double reading with two radiologists in detecting breast cancer. In fact, this reading combination resulted in a 4% increase in screen-detected cancers. The study also found that the consensus discussion was effective in ensuring that the higher abnormal interpretation rate for AI plus one radiologist did not translate into an increased recall rate.

In 2023, the results of another prospective study evaluating the performance of a commercially available AI software for breast cancer detection by Ng et al. were published [83]. The AI system was used as an additional reader in a standard double-reading process. According to this study, the AI-assisted reading process could increase the detection rate by 0.7–1.6 per 1000 cases, incrementing recalls by 0.16–0.30% with 0–0.23% extra unnecessary recalls. Overall, subsequent human assessment of AI-flagged examinations allowed a 0.1–1.9% increase in positive predictive value (PPV). Notably, most detected neoplasms were invasive and small-sized (≤10 mm), demonstrating the potential of AI-assisted reading to improve the early detection of prognostically relevant breast lesions with a minimal number of additional unnecessary recalls.

In late 2022, the Radiological Society of North America (RSNA) organized a public breast cancer detection challenge attended by over 1600 teams [68]. Notably, participants had to publicly release the source code of their model to allow external evaluators to reproduce their achievements. The results of the challenge have been published in May 2023 [91]. The 1st place solution employed a multi-step pipeline [92], including a YOLOX-nano detector to extract the breast ROI [93], several preprocessing operations to optimize the input images, and a fourfold ConvNext-small network for final classification [18], achieving an AUC = 0.93.

Design information, key results, and highlighted limitations of the main studies evaluating deep learning applied to conventional mammography have been summarized in Table 4.

### 5.2. Digital Breast Tomosynthesis

A 2023 review article by Magni et al. explored the role of AI and deep learning in the interpretation of digital breast tomosynthesis (DBT) images [94]. According to the authors, studies have shown that AI-assisted DBT interpretation can increase sensitivity, reduce recall rates, and decrease overall double reading workload by producing selective synthetic images. However, larger retrospective and prospective studies will be required to assess the benefits of AI-based DBT interpretation when compared to AI-based FFDM reading, the latter potentially able to reduce the diagnostic gap to tomosynthesis thanks to the higher sensitivity of AI-assisted radiologists.

One of the largest and most real-world-representative retrospective evaluations of AI-based DBT interpretation was published in December 2021 by Romero-Martín et al. [84]. A total of 15,999 DBT and digital mammography (DM) examinations were retrospectively collected and independently scored by an AI system. AI achieved an AUC = 0.93 for DM and 0.94 for DBT. For DM, AI demonstrated noninferior sensitivity as a single or double reader, with a reduction in recall rate of up to 2%. For DBT, AI demonstrated noninferior sensitivity as a single or double reader but with a higher recall rate of up to 12.3%.

### 5.3. Contrast-Enhanced Mammography

The potential role of AI in the interpretation of contrast-enhanced mammography examinations was recently pointed out in a commentary by Zhang et al. [95]. As mentioned in a 2024 review article by Kinkar et al. [96], several studies demonstrated good performance from automatic segmentation and classification models for contrast-enhanced mammograms.

In April 2023, Zheng et al. published the results of a prospective, multicenter study assessing deep learning-based breast cancer segmentation performed on contrast-enhanced mammography (CEM) examinations [85]. A total of over 1900 Chinese women with single-mass breast lesions on CEM images before biopsy or surgery were included in the study. A fully automated pipeline system performed the segmentation and classification of breast lesions, achieving a DSC = 0.837 ± 0.132 and an AUC = 0.891, respectively.

Another retrospective study published in June 2023 by Beuque et al. aimed to evaluate a machine learning-based tool able to identify, segment, and classify breast lesions in CEM images from recalled patients [86]. Low-energy and recombined images were preprocessed, manually segmented, and subsequently used to train deep learning models for automatic detection, segmentation, and classification. The detection and segmentation model achieved a mean DSC = 0.71 and 0.80 at the image and patient level, respectively. A handcrafted radiomics classifier was also trained to assess both manually and automatically segmented lesions. The two models combined achieved the best classification performance, measured as an AUC = 0.88 when interpreting manual segmentations and 0.95 when using automatically generated segmentations.

In a retrospective study published in August 2023 [87], Qian et al. trained and assessed a classifier based on a multi-feature fusion network architecture, using both dual-energy subtracted (DES) and low-energy (LE) bilateral, dual view images as input data. The classifier achieved good diagnostic performance with an AUC = 0.92 on an external CEM dataset, while retaining adequate performance when tested against an external FFDM dataset.

### 5.4. Ultrasound

A review article published in early 2024 by Dan et al. examined the applications and performance of deep learning-based breast ultrasound (BUS) evaluation [97]. Overall, when compared to screening mammography, studies were fewer, smaller-sampled, and more heterogeneous in their methodology and results.

When considering diagnostic detection, segmentation, and classification tasks, both standalone and human-assisting AI were evaluated, observing diverse results but mostly without significant superiority when compared to human readers. Some studies reported a slight improvement in diagnostic performance when pairing AI with inexperienced radiologists. Overall, both standalone and assistive DL-based systems seemed slightly more specific than average human readers, while their sensitivity remains unclear [97].

In a prospective study published in late 2022, Gu et al. developed a DL-based classifier for BUS, assessing its performance on a large multicenter dataset comprising images from over 5000 patients. The standalone model demonstrated good performance, achieving an AUC = 0.913, comparable to experienced radiologists and significantly higher than inexperienced radiologists [88]. Moreover, AI assistance improved the accuracy and specificity of radiologists without altering sensitivity.

A 2020 study by Zheng et al. assessed the performance of a deep learning-based radiomics model for preoperative US classification of axillary lymph nodes (ALNs) in patients with early-stage breast cancer, achieving good diagnostic and metastatic burden prediction performance with AUC = 0.902 and 0.905, respectively [98]. A subsequent study with a public model implementation by Sun et al. achieved a lower performance, with an AUC = 0.72 in the testing dataset [99].

Lyu et al. developed a segmentation model for breast lesions on ultrasound examinations using an attention module to enhance edge and detail recognition [100]. The model demonstrated promising performance with a DSC = 0.8 on external datasets, proving the ability of DL-based models to automate lesion segmentation on ultrasound images.

### 5.5. Magnetic Resonance Imaging

A review article published in July 2023 by Adam et al. explored deep learning applications for breast cancer detection by MRI [101]. CNNs have been trained and used for classification, object detection and segmentation tasks achieving good performance in small-sampled studies. However, much like with ultrasound examinations, large prospective and retrospective studies properly assessing deep learning performance of AI-based breast MRI interpretation in real world scenarios are still lacking.

Several recent studies explored AI-based automatic segmentation of breast cancer on dynamic contrast-enhanced MRI (DCE-MRI) examinations. This technique allows precise and almost effortless segmentation of neoplastic lesions, enabling complete volume calculation for staging, treatment planning, and response evaluation while also offering an optimal data source for radiomic analyses.

In a study published in February 2023 by Janse et al. [89], an nnU-Net segmentation pipeline was trained to segment locally advanced breast cancer (LABC), assessing neoadjuvant chemotherapy response by residual cancer volume estimation. Manually segmented ground-truth data from 102 LABC patients was used to train the model, and an independent testing cohort consisting of 55 LABC patients from four institutions was used for performance evaluation. Automated segmentation resulted in a median DSC = 0.87. Automated volumetric measurements were significantly correlated with functional tumor volume (FTV). Notably, pre-trained model weights were published by the authors to allow local reproduction of the results [102].

Another notable subset of deep learning applications for breast MRI has been the generation of synthetic post-contrast images from pre-contrast sequences using GANs. While still in its infancy, this technique might prove crucial in future years to improve breast cancer staging and treatment response evaluation in patients who are not eligible for intravenous contrast administration.

In an article originally published in November 2022, Chung et al. investigated the feasibility and accuracy of generating simulated contrast-enhanced T1-weighted breast MRI scans from pre-contrast MRI sequences in biopsy-proven invasive breast cancer using a deep neural network [103]. Synthetic images were qualitatively judged by four experienced breast radiologists and quantitatively assessed using indexes such as the DSC. Most of the simulated scans were considered diagnostic quality, and quantitative analysis demonstrated strong enhancing tumor similarity with a DSC = 0.75 ± 0.25.

A late 2023 study published by Osuala et al. explored the feasibility of producing synthetic contrast-enhanced images by translating pre-contrast T1-weighted fat-saturated breast MRI to their corresponding first DCE-MRI sequence using a GAN [104]. The authors subsequently used a nnU-Net pipeline to assess segmentation performance on synthetic post-contrast images. While the cancer segmentation performance expressed as DSC was indeed lower on synthetic post-contrast images when compared to actual DCE-MRI, this study highlighted the potential of this technique, warranting further studies to improve the GAN design and training approach. The complete source code for the deep learning models used in the study was publicly released [105].

A recently developing field of deep learning applications employing breast magnetic resonance imaging is the prediction of crucial prognostic information, such as neoadjuvant chemotherapy response, through radiomic analyses. Older radiomic approaches exploited manual or semi-automated selection and extraction of hard-coded features from imaging, pathology, and clinical data. Modern deep learning models effectively automate feature extraction, significantly streamlining model development and potentially delivering better results [27].

In early 2023, Li et al. published the results of a retrospective study evaluating the performance of a deep learning-based radiomic (DLR) model for predicting pathological complete response (pCR) to neoadjuvant chemotherapy in breast cancer [90]. Two conventional, handcrafted feature extraction-based radiomic signature models from different treatment periods were also prepared for comparison. The performance of the DLR model combining pre- and early treatment information from DCE-MRI lesion segmentation was better than both the radiomic signature models (AUC = 0.900 vs. 0.644 and 0.888, respectively). The combined model, including pre- and early treatment information and clinical characteristics, showed the best ability with an AUC = 0.925, demonstrating a valuable role in predicting treatment response rates.

## 6. Deep Learning in Breast Cancer Imaging: Novel Techniques

### 6.1. Thermography

Thermography is a non-ionizing imaging modality that measures heat patterns on the surface of the skin overlying the breast tissue to detect abnormal thermal areas potentially indicating malignant lesions [106]. The idea behind thermography is that breast tumors tend to generate different heat patterns when compared to surrounding normal tissue due to altered blood vessel growth and metabolic activity.

A thermographic breast examination involves the acquisition of static and dynamic images for each breast from multiple angles using an infrared camera. Images are then analyzed for abnormalities such as focal hotspots, asymmetric temperature distributions, and altered vascular patterns.

One of the main advantages of thermography is that it does not use ionizing radiation, which can be a concern, particularly for women in younger age groups needing regular screenings or undergoing pregnancy and lactation. Additionally, thermography is non-contact; therefore, it does not require uncomfortable compression of the breast tissue. Moreover, it is less expensive than MG, making it easily implementable in lower-income regions and lower-level healthcare points.

Thermography may also be able to detect changes in the breast tissue at an earlier stage than mammography by recognizing infra-radiological abnormalities related to increased blood flow and metabolic activity [107].

Many research works have explored deep learning models to detect, segment, and classify breast cancer in thermographic images with promising results.

In 2017, Mambou et al. developed a CNN classifier to detect the presence of breast cancer on thermograms coupled with a conventional ML classifier for assessing uncertain DL outputs [108]. The model was trained and tested against a dataset of 67 subjects, including 43 normal subjects and 24 positive patients, achieving complete classification accuracy.

In a 2021 publication, Mohammed et al. compared the thermogram classification performance of three models based on the popular Inception architecture, introducing a modified variant of InceptionV4 (MV4) capable of 7% faster inference when compared to the original [109]. Both the original and modified InceptionV4-based models achieved almost complete classification accuracy.

In 2022, Alshehri et al. used attention mechanisms (AMs) to improve the detection performance of a CNN-based thermogram classifier, achieving up to 99.46% accuracy versus 92.3% for a CNN without AM [110]. In a subsequent early 2023 study, the same authors achieved up to 99.8% accuracy by using a deeper CNN architecture coupled with AMs [111].

In a 2022 article, Mohamed et al. described the development of a fully DL-based pipeline for breast cancer detection on thermograms, combining a U-Net breast tissue segmentation model from thermograms with a bespoke classifier [112]. The pipeline achieved 99.3% accuracy. Notably, an adapted VGG16-based classifier included for comparison was able to achieve complete accuracy.

In 2023, Civiliban et al. designed a Mask R-CNN-based model with a ResNet-50 backbone capable to detect and segment breast lesions on thermograms, delivering excellent performance with a 0.921 mean average precision for detection and a 0.868 overlap score for segmentation [113].

In early 2024, Khomsi et al. reported the implementation of a custom feed-forward neural network for estimating tumor size based on thermographic data [114].

Despite promising research results, few studies have evaluated the role of DL models for thermography in real-world clinical scenarios.

In late 2021, Singh et al. first published the results of a multicentric study conducted on 258 symptomatic patients undergoing thermography followed by MG and/or US, evaluating the performance of a commercial AI-based thermal breast screening device [115]. Promising results were observed, with the platform delivering an AUC = 0.845 with slightly lower sensitivity (82.5% vs. 92%) when compared to MG.

In 2023, Bansal et al. reported the results of a prospective study conducted between 2018 and 2020 on 459 women, both symptomatic and asymptomatic, evaluating the same device [116]. Thermography was followed by MG and other diagnostic modalities to confirm the findings. The device demonstrated non-inferior performance when compared to MG while delivering better sensitivity in women with dense breasts.

### 6.2. Microwave Breast Imaging

Microwave breast imaging (MBI) is a relatively new technology that is being explored as an alternative or complement to traditional methods of breast cancer screening, such as mammography. The basic idea behind microwave imaging is to use low-power radio waves to create images of the breast tissue, identifying areas with different dielectric properties [117].

During a microwave imaging examination, the patient sits with her breasts placed on a desktop-like unit. A set of antennas emit and receive very low-power microwaves that penetrate the breast tissue and measure the reflection and scattering of these waves by the tissues inside. This information is then used to construct an image of the breast tissue, which can be analyzed for signs of abnormalities such as tumors.

Unlike mammography and much like thermography, microwave imaging does not use radiation and does not lose as much performance with dense breasts, making it possible to extend screening programs to younger women and to increase examination frequency, thereby potentially improving the diagnosis of early-onset breast cancer and reducing the occurrence of interval cancers. Moreover, since there is no compression of the breast during a microwave imaging exam, many women find it more comfortable than mammography. Finally, microwave imaging systems are generally less expensive to build and maintain than mammography equipment, which could make them a more accessible option in underserved areas.

Overall, while microwave imaging shows promise as a tool for breast cancer screening, further research is needed to establish its effectiveness and safety compared to mammography. Currently, microwave imaging is not widely available and is primarily used in clinical trials and research settings. Moreover, at present time microwave imaging seems to have some performance limitations compared to mammography. A recent multicentric, single-arm, prospective, stratified clinical investigation evaluated a commercial microwave imaging system’s ability to detect breast lesions, demonstrating overall worse accuracy when compared to a reference standard (73%) [118].

In early 2022, Moloney et al. published the results of the first clinical study investigating the performance of a commercial microwave breast imaging system capable of automated lesion detection and characterization via physical lesion features [119]. Among the 24 symptomatic patients included, the MBI system correctly detected and localized 12 of 13 benign lesions and 9 out of the 11 cancers, including a radiographically occult invasive lobular neoplasm. Further technical details regarding the automated classification algorithm were published in a separate article [120].

At least two commercial solutions from Italy [121], and France [122], respectively, have been made available and are undergoing evaluation in healthcare facilities. A larger scale, European-funded prospective clinical study involving 10,000 patients across 10 centers is currently ongoing and will end in November 2026 [123].

Despite initial research efforts with promising results [124], no studies have yet explored the role of deep learning applied to MBI in clinical and screening scenarios.

### 6.3. Other Techniques

Breast elastography is a non-invasive, ultrasound-based technique used to evaluate the stiffness or elasticity of breast tissue. It can be used in conjunction with other modalities to help distinguish between benign and malignant breast lesions, as cancerous tumors tend to be stiffer than surrounding healthy tissue [125]. In 2015, Zhang et al. first developed a simple two-layer deep neural network for feature extraction on shear-wave breast elastography images, combined with a conventional machine learning algorithm for malignancy prediction, showing better classification performance than models based on handcrafted features [126]. Recent research works using fully deep learning-based models confirmed the role of AI in automating evaluation, reducing inter-observer variability, and increasing the interpretation accuracy of inexperienced radiologists [127].

Breast-specific gamma imaging (BSGI) is a nuclear medicine study that involves injecting a patient with a radioactive tracer and then using a special camera to create images of the breast tissue [128]. This technique can help detect small tumors that may not be visible in other types of imaging studies [129]. Yu et al. recently assessed the performance of a ResNet18-based classifier for BSGI images with positive results [130].

Positron emission mammography (PEM) is another nuclear medicine study that involves injecting a patient with a radioactive tracer and then using a specialized scanner to create detailed images of the breast tissue [131]. PEM can be helpful in identifying early-stage breast cancers and determining whether a lump detected by other methods is benign or malignant [132]. At present, no significant studies regarding DL-based classification or segmentation of breast cancer on PEM have yet been conducted.

Optical imaging techniques use light to visualize the breast tissue, often by shining near-infrared light through the skin [133]. These techniques can provide information about blood flow and oxygenation levels within the breast tissue, which can help distinguish between healthy and cancerous tissues in a screening setting [134]. Zhang et al. developed a fusion optical tomography–ultrasound DL model for breast cancer classification, achieving competitive performance with an AUC = 0.931 [135].

Key potential advantages and current challenges for novel breast cancer imaging techniques have been summarized in Table 5, while the most prominent studies on this topic have been listed In Table 6.

## 7. Deep Learning in Breast Cancer Imaging: Recent Advancements and Trends

Figure 6 represents the principal novelties in deep learning applied to medical imaging that have impacted recent AI-assisted breast cancer imaging research. Technical advancements and new study design trends will be discussed separately in the next sections.

### 7.1. Technical Advancements

#### 7.1.1. Vision Transformers

Vision transformers (ViTs) are a type of neural network architecture that has recently been used for CV tasks such as image classification as an alternative to CNNs. Introduced in 2020 by Dosovitskiy et al. [136], they have been gaining popularity due to their strong performance on various benchmarks.

ViTs treat an image as a sequence of non-overlapping patches and feed these patches into a transformer model, an architecture originally developed for natural language processing tasks. The transformer processes each patch independently and learns to attend to different parts of the input sequence to make predictions. By treating the image as a sequence rather than a grid of pixels, ViTs can capture long-range dependencies and global context that may be missed by CNNs.

ViTs have been shown to achieve state-of-the-art performance on several image classification datasets, including ImageNet [137]. However, they also require more computational resources than CNNs, particularly when dealing with large image resolutions. Additionally, because ViTs do not explicitly encode any notion of position or spatial relationship, they may struggle to capture certain types of information that are easily captured by CNNs. Nonetheless, ViTs represent an exciting new direction in computer vision research and have opened many avenues for further investigation.

ViTs have recently started to gain attention in the medical imaging field, promising to improve diagnostic accuracy and efficiency [138]. One potential advantage of using ViTs in this realm is their ability to capture long-range dependencies and global context. Unlike CNNs, which focus primarily on local patterns, transformers can identify correlations across distant regions of an image, potentially revealing important clinical findings that might otherwise go unnoticed. Moreover, transformers can handle variable-sized inputs, allowing them to analyze medical images of varying shapes and sizes without requiring preprocessing steps like cropping or resizing.

Several studies have demonstrated ViTs’ potential in medical imaging applications. For example, a 2021 study showed that a vision transformer could accurately detect COVID-19 from chest computed tomography (CT) scans, outperforming established CNN architectures [139]. Other recent works have explored the use of vision transformers in analyzing mammograms [140], brain MRI scans [141], and many more medical images.

Despite the promising early results, vision transformers still face some challenges in practical medical imaging scenarios. One issue is their relatively high computational cost, which could limit their adoption in resource-constrained settings. Furthermore, vision transformers usually require larger amounts of labeled data to train effectively, a problematic factor given the relatively limited availability of annotated medical images.

To date, studies implementing ViTs for breast cancer classification are still relatively few when compared to those evaluating CNNs. Recently, Ayana et al. developed a ViT-based mammography classifier adapted by transfer learning from a pre-trained model, which was able to achieve better performance than state-of-the-art CNN-based classifiers [142]. However, in a comparison study published in late 2023, Cantone et al. observed that ViT-based models may deliver worse mammography classification performance than CNNs when trained with small datasets [143].

#### 7.1.2. Improved Convolutional Neural Networks: ConvNeXt

ConvNeXt is a neural network architecture that takes inspiration from traditional ResNet models but introduces new components to improve their performance [18].

The key idea behind this architecture is to apply modern design principles used in transformer-based architectures (such as Vision Transformers) to CNNs. This includes features such as larger kernel sizes to improve contextual understanding, depthwise separable convolutions to reduce computational complexity, layer normalization instead of batch normalization to improve training and generalizability, and spatial resolution reduction via strided convolutions rather than pooling layers to reduce information loss during downsampling.

One significant advantage of this architecture over other state-of-the-art models, especially when compared to Visual Transformers, is its simplicity and computational efficiency while achieving comparable or even better accuracy on various benchmarks [18].

ConvNeXt v2 was designed to address several limitations of the original model [144]. Some of the changes include improved layer normalization, dynamic depth, better activation functions, and more efficient spatial reduction.

Overall, ConvNeXt has become one of the most competitive computer vision classification models currently available. However, despite being less complex than ViTs, it is still more computationally expensive and harder to train than CNNs, while not always delivering significantly better performance [143].

In a 2022 study by Hassanien et al., a ConvNeXt-based classifier was used to successfully predict breast tumor malignancy on ultrasound images, outperforming other popular CNN- and ViT-based models [145]. A ConvNeXt-based classifier was also used in the winning entry for the RSNA Screening Mammography Breast Cancer Detection Challenge [68], achieving an AUC = 0.93 for the malignancy classification task [92].

#### 7.1.3. New Object Detectors: The YOLO Series

YOLO is a real-time object detection architecture that treats this task as a regression problem rather than a typical two-step process of first identifying regions of interest and then classifying those regions [19]. This approach makes it faster and more efficient than traditional methods such as R-CNN or Fast R-CNN while retaining high accuracy with its latest iterations.

Several variations of the YOLO architecture have been released by different developers, each with incremental improvements or specific target applications. Notably, Wang et al. have focused on overall detection accuracy rather than absolute speed, publishing YOLOv7 in 2022 [146], and YOLOv9 in early 2024 [147]. YOLOv8, a variation of the v7 model developed by a third-party company, has also become widely popular in the computer vision field due to its ease of implementation [148].

In early 2021, Aly et al. assessed the performance of a YOLO-based breast mass detector and classifier for FFDMs, demonstrating superiority to other conventional CNN architectures [149]. In a 2022 study, Su et al. combined a YOLOv5 model with a local-global transformer architecture (LOGO) to detect and segment breast masses on mammograms, achieving good performance on the CBIS-DDSM and INBreast datasets [150]. Hassan et al. combined a YOLOv4 detector with a ViT classifier to recognize and characterize breast lesions on FFDM and CEM images, demonstrating state-of-the-art performance [151]. Prinzi et al. recently compared the performance of different YOLO-based models for breast cancer detection in mammograms [152]. The authors analyzed not only bounding box results but also saliency maps to evaluate model activation in local image areas, highlighting the importance of AI explainability to provide radiologists and clinicians with better insights about the model’s predictions.

#### 7.1.4. Automated Segmentation Pipelines: nnU-Net

Automated segmentation pipelines are machine learning models that can automatically identify and segment structures or regions of interest in medical images. These pipelines have gained significant attention in recent years due to their potential to improve the efficiency and accuracy of image analysis tasks in various clinical applications, including tumor detection, organ segmentation, and disease diagnosis.

nnU-Net is a popular open-source automated segmentation pipeline that has shown state-of-the-art performance in several medical imaging challenges [33].

The key advantage of nnU-Net over other segmentation workflows is its ability to learn optimal network architectures for different datasets without requiring extensive manual tuning. This is achieved through a nested and unified U-Net architecture, which allows the model to adaptively adjust its depth and width based on the complexity of the input data. Additionally, nnU-Net includes pre-processing steps and post-processing techniques to further enhance the quality of the predicted segmentation. Moreover, it utilizes advanced augmentation techniques during training to increase the robustness of the model to variations in image acquisition parameters.

Released in early 2023, nnU-Net V2 represents a complete overhaul of the pipeline [153]. Despite delivering the same segmentation performance, it features development and usability improvements, compatibility with additional input file formats, hierarchical labels support and optimizations for more hardware and software platforms.

Overall, nnU-Net represents a promising approach for automating medical image segmentation tasks, delivering state-of-the-art segmentation performance and allowing even non-technically experienced developers to create high quality models, thanks to its accuracy and ease of implementation.

nnU-Net has been successfully used for a wide variety of medical imaging segmentation tasks, including breast imaging studies by Janse et al. [89], and Osuala at al. [104], and many more applications can be expected in future researches.

#### 7.1.5. Deep Learning-Based Radiomics Classifiers

Deep learning-based radiomics uses deep neural networks to analyze images, making predictions about clinical features and outcomes, such as histopathological characteristics, prognosis, or treatment response [27]. Deep learning algorithms are particularly well-suited for analyzing large, high-dimensional datasets like those generated by medical imaging techniques. This approach is opposed to conventional, handcrafted radiomics, which manually extracts quantitative features from medical images using predefined mathematical formulae and then evaluates them using statistical machine learning algorithms.

One key advantage of deep learning-based radiomics classifiers is their ability to learn hierarchical representations of data, where lower-level features are combined to form more abstract, higher-level features. This allows the models to automatically identify complex patterns in the data that may not be apparent through manual feature engineering. Additionally, deep learning models can handle noisy or missing data better than traditional machine learning algorithms, making them more robust to variations in image quality or acquisition protocols.

However, there are also several challenges associated with developing and implementing deep learning-based radiomics classifiers in clinical practice. One major consideration is the need for large, diverse training datasets that accurately represent the patient population of interest. Another challenge is ensuring the reproducibility and generalizability of the models across different imaging platforms, acquisition parameters, and patient cohorts. Explainability of the models’ predictions is also an issue since neural networks effectively behave like black boxes, outputting the result of the training and inference process but providing little insight about the features they rely upon to make judgments [154]. Finally, rigorous validation and testing of the models are needed to ensure their accuracy and reliability in real-world settings.

Examples of DL-based radiomics classifiers applied to breast cancer imaging include algorithms for tumor [145] and lymph node malignancy assessment [98], pathologic markers evaluation [155], and treatment response prediction [90].

### 7.2. Study Design Trends

#### 7.2.1. Prospective versus Retrospective Approach

Prospective studies and retrospective studies are two types of observational research designs used in epidemiology and other fields to investigate the causes and outcomes of diseases or health-related events [156].

In a prospective study, researchers collect data specifically for the purpose of the study, which allows for more control over the variables being measured and reduces the risk of bias. By contrast, in a retrospective study, researchers rely on pre-existing records or databases that may not have been collected with the same level of detail or consistency.

Because participants are enrolled at the beginning of a prospective study, there is less chance of selection bias affecting the results. In contrast, retrospective studies may be subject to selection bias if certain groups are more likely to be represented [157]. In the cancer imaging field, selection bias may derive from using cancer-enriched datasets [77], which do not reflect the true prevalence of neoplastic disease in the general population and could therefore skew classification performance results.

Moreover, prospective studies typically involve longer follow-up periods, allowing researchers to observe outcomes over a longer time and capture more detailed information on delayed events, such as interval cancers or next-round screen-detected cancers.

Finally, prospective studies often include extensive assessments of participants’ clinical and histopathological characteristics, helping to identify additional associations and trends, while retrospective studies may have limited information on these factors.

However, prospective studies also have disadvantages, including higher costs, longer timelines, and potential attrition bias [158].

Historically, most studies regarding deep learning in the cancer imaging field have followed a retrospective approach to satisfy the need for quick evaluation of novel deep learning-based models. However, with the increased availability of high-quality models and regulatory-approved solutions, several prospective studies have also been conducted.

Recent prospective studies evaluating deep learning-based models applied to breast cancer imaging include works by Dembrower et al. [82], Ng et al. [83], Zheng et al. [85], and Gu et al. [88], whose key benefit lies in the ability to more closely reproduce a common clinical scenario in terms of disease prevalence, information availability, and interpretation setting.

#### 7.2.2. AI Integration Strategies

AI integration into medical imaging examinations takes various forms, each with distinct benefits and drawbacks. One way is using standalone classification systems that process raw data and generate reports independently. AI-based triage systems that only trigger radiologist intervention when a critical predicted abnormality threshold is reached have been tested with successful results [81]. Although this strategy offers quick turnaround times, potential cost savings, and improved consistency, it may lack adaptability in non-standard scenarios and, most importantly, raises ethical questions concerning explainability and accountability [159].

Another approach employs AI as an assistant to a single radiologist during image interpretation (AI-assisted single reading). This method combines the efficacy of deep learning with human expertise, enhancing diagnostic performance while potentially reducing cognitive strain on individual practitioners. It also provides educational value for trainees and early career professionals, helping to minimize interobserver variability among radiologists. However, successful implementation requires seamless integration between AI systems and PACS, increased upfront costs, the establishment of trust between radiologists and AI models, and avoidance of complacency or underestimation of residual errors introduced by AI assistance [160].

A third strategy involves two radiologists reviewing the same case individually, subsequently supported by an AI system (AI-assisted double reading, also defined as *triple reading*). Following separate assessments, they compare conclusions, aiming to achieve higher diagnostic precision in terms of sensitivity and specificity. This approach detects additional inconsistencies that serve as learning opportunities, enables objective evaluation of AI performance against human experts, and stimulates competition between conventional techniques and novel technologies. Nevertheless, this method consumes substantial resources due to added staffing needs, lengthens reporting times, necessitates consistent collaboration across cases, and demands careful consideration when allocating roles and responsibilities.

The 2023 prospective study by Dembrower et al. represents the most complete comparative evaluation of the three strategies applied to breast screening examinations [82]. Predictably, AI-assisted double reading delivered the highest number of abnormal interpretations, followed by assisted single reading and unassisted double reading, respectively. Conversely, standalone AI reading produced the lowest number of abnormal interpretations and therefore the lowest recall rate. However, the cancer detection rate was similar for all the strategies, demonstrating a potentially higher specificity for AI as a single reader without significant sensitivity loss.

Despite AI single reading not being realistically implementable in the short term due to ethics and liability issues with current regulations, its results highlight the diagnostic accuracy achieved by modern breast cancer detectors and warrant future research both in the clinical and bioethical field to assess its future possible applications.

On the other hand, when considering the implementation of AI-assisted human reading strategies, more studies will be needed to estimate the impact of automation bias on radiologists’ performance and develop strategies to address or reduce this issue [160].

#### 7.2.3. Public Challenges

Public challenges constitute a recent trend in the deep learning research and development scene and can play a critical role in driving innovation and progress in the field of deep learning applied to medical imaging [161]. One key advantage of these challenges is that they provide a standardized benchmark for comparing different methods and establishing best practices. By using identical datasets and evaluation criteria, researchers can more easily assess the relative strengths and weaknesses of various algorithms and techniques.

In addition to promoting benchmarking and comparison, public challenges can also foster greater collaboration and community building among researchers. These events attract participants from a wide range of academic and industrial backgrounds, creating opportunities for interdisciplinary dialogue and knowledge exchange. Moreover, by tackling real-world problems and datasets, public challenges can help ensure that research remains relevant and applicable to clinical settings.

Another benefit of public challenges is that they facilitate data sharing and access, providing participants with high-quality curated datasets that may otherwise be difficult to obtain. This not only enables fairer evaluations of competing methods but also encourages further exploration and analysis of the data.

Furthermore, public challenges can promote transparency and reproducibility in research through their emphasis on documentation and openness. Encouraging participants to share their code and methodologies makes it easier for others to evaluate and build upon their work, ultimately contributing to more robust and trustworthy scientific findings.

Finally, public challenges offer motivational benefits for researchers, who may be drawn to compete for prizes, recognition, or simply the satisfaction of solving challenging problems.

Examples of public medical imaging challenges include initiatives from the Radiological Society of North America [162], the Medical Image Computing and Computer-Assisted Intervention (MICCAI) Society [163], and the Grand Challenge project [164].

Overall, public challenges have been and will represent a powerful tool for advancing research in deep learning applied to medical imaging, offering numerous social, technical, and intellectual benefits for all involved disciplines.

## 8. Discussion

### 8.1. State of the Art

The status of key DL applications for breast cancer imaging are highlighted in Figure 7.

Multiple neural network architectures have been applied in this field, but most of the established implementations involve CNNs for image interpretation. CNNs have been successfully used for medical imaging-related computer vision tasks such as classification, object detection, and segmentation. Investigational studies have employed GANs for synthetic image generation and LLMs for automated report production.

The creation of a deep learning model typically involves the development of a neural network pipeline and the collection of an adequate dataset for its training. Either high-performance hardware or external servers are required to perform training and inference with reasonable speed. The quality of the model can be evaluated with standardized performance metrics such as AUC for classification and DSC for segmentation. AI predictions should be easily accessible and understandable by radiologists and clinicians via platforms integrated into PACS.

Several curated datasets have been publicly released for different breast imaging modalities, but most of them contain mammograms, while only a few include ultrasound and magnetic resonance images. These datasets usually provide ground truth annotations from experienced radiologists, either as image labeling or lesion identification, with some of them also reporting relevant clinical and histopathological information. Larger and higher quality datasets have also been made commercially available, but costs constitute a significant adoption barrier.

Breast cancer screening using conventional mammography has been one of the most researched and well-established applications of deep learning-based medical imaging models. Several AI-based detection platforms have received approval from international regulators during the last few years, and multiple studies have positively assessed their performance in controlled environments. Overall, studies have shown remarkable performance for many of these models, highlighting their potential to increase the accuracy and efficiency of mammography interpretation while reducing false positive and false negative rates, inter-observer variability, and radiologists’ workload. Common study limitations include unknown reproducibility of results across different healthcare facilities due to different demographics and scanning devices, use of retrospective designs, cancer-enriched datasets, and other inclusion criteria potentially skewing results due to selection bias, and funding by AI software companies. Recently published prospective studies including a larger number of patients have helped better represent AI performance and impact in real screening scenarios.

Comparatively, fewer studies have involved other commonly used breast imaging modalities such as digital breast tomosynthesis, contrast-enhanced mammography, ultrasound, and dynamic contrast-enhanced magnetic resonance imaging. Specifically, in addition to study limitations previously cited with respect to mammography-applied AI research, small training and testing sample sizes imply significant constraints in terms of scalability and generalizability. Overall, results have been more heterogeneous and inconsistent across studies, and current AI applications for these modalities mostly remain in an investigational state. Still, interesting applications of deep learning based in this field include automated segmentation for quick and precise cancer burden assessment, assistance for inexperience radiologists to improve human accuracy and reduce inter-observer variability, generation of synthetic images for artificial contrast-enhancement and long studies summarization, and advanced characterization of the primary tumor and lymph nodes for prognostic and predictive analyses.

To date, novel and ancillary techniques such as thermography, microwave-based imaging, elastography, breast-specific gamma imaging, positron emission mammography, and optical imaging have played a limited role in the diagnosis and management of breast cancer patients. Deep learning might help to enhance the strengths and reduce the weaknesses of these modalities, potentially paving the way for their future introduction in clinical practice as alternative diagnostic approaches for specific contexts and subsets of patients. Nevertheless, research in this area is still lacking, overall limited in numbers, and mostly confined to investigational settings.

Multiple technical advancements have been made in the computer vision field during the last few years. ViTs are novel neural networks for computer vision tasks that have been proposed as an alternative to CNNs, adopting an LLM-like architecture and achieving state-of-the-art performance on common CV benchmarks despite requiring bigger datasets and more potent computing resources for effective training. Modern CNNs, such as ConvNeXt, represent a bridge between conventional CNNs and ViTs, providing excellent performance with lower complexity than transformer-based models. New object detection and segmentation pipelines, such as the YOLO family and nnU-Net, provide developers with easy-to-use tools to create high-quality models. Radiomics, an emerging field that traditionally involved the extraction of quantitative features from medical images using conventional machine learning techniques to predict clinical and histopathological information, has recently been revolutionized by DL-based models able to extrapolate hidden properties and infer complex correlations, improving patient outcomes prediction and tumor characteristics assessment, and ultimately increasing the potential for treatment personalization.

New study designs recently adopted in the field, such as prospective approaches comparing different integration strategies, have provided novel and more accurate insights about the effects of AI implementation into clinical practice, including the complex interaction between radiologists and software assistants and its impact on human performance. Public challenges have greatly contributed to deep learning research by providing a common ground for developers to train and test their models against and enforcing transparency and reproducibility of experimental results.

### 8.2. Limitations, Challenges and Future Directions

Despite the milestones achieved throughout recent years, several critical limitations and challenges remain to be addressed for the successful integration of AI-based systems into clinical practice. Figure 8 provides a schematic representation of these issues.

#### 8.2.1. Generalizability

Medical imaging data comes with significant variability across different centers due to factors such as imaging protocols, equipment specifications, and patient demographics. These variations can introduce biases and hinder model generalizability [165]. Addressing this challenge will require efforts to standardize data collection protocols and collaboration across institutions to provide high-quality, representative datasets for DL training. Moreover, the affordability of these datasets should be promoted to extend their availability to as many worldwide researchers as possible. Further studies will be needed to assess the consistency of AI performance across different clinical settings and to evaluate the role of personalized (re)training strategies.

#### 8.2.2. Multimodal Interpretation

Experienced radiologists usually consider multiple elements when assessing imaging examinations, such as previous studies, results from different modalities, lab tests, pathological specimens, and clinical status. Currently, AI-based interpretation of multimodal healthcare data poses significant challenges due to the complexity of integrating information from different sources in a single algorithm. DL models capable of handling multimodal data effectively will be needed to fully leverage the complementary information provided by various techniques [166].

#### 8.2.3. Costs

Software costs and hardware requirements for deep learning inference and training constitute a significant adoption barrier. High-performance computing resources are essential for training complex models and processing large volumes of medical imaging data. Access to such resources can be costly and may limit the scalability of deep learning applications in clinical settings [167]. Development of open training and inference platforms for medical imaging AI models, along with more efficient and cost-effective hardware solutions, might prove key to extending clinical adoption [168].

#### 8.2.4. Privacy

The use of external servers to offload inference operations presents concerns regarding privacy and security. Transmitting sensitive medical information to external servers raises potential risks of data breaches and compromised patient confidentiality. Implementing secure and privacy-preserving solutions for data processing while maintaining computational efficiency will be essential for ensuring the widespread adoption of deep learning in medical imaging [55].

#### 8.2.5. Human–AI Interaction

The introduction of deep learning-based assistance may induce excessive reliance on AI models by radiologists, potentially leading to complacency or overconfidence in automated diagnoses, also known as automation bias [160]. Radiologists must remain vigilant and critically evaluate model outputs, integrating them with clinical expertise to ensure accurate interpretations. The interaction between healthcare professionals and AI-based algorithms is a new and relatively unexplored field warranting future research for its complex implications on human performance.

#### 8.2.6. Explainability, Ethics, and Liability

The architecture of deep neural networks effectively makes them black boxes that convert input data into output predictions without providing direct insights as to why a certain prediction was made. This element implies complex ethical and legal considerations in hypothetical future scenarios where the diagnosis and management of diseases will be strongly influenced by AI models, either with or without the direct involvement of clinicians. Efforts to increase model explainability and interpretability, collectively defined as Explainable Artificial Intelligence (XAI) [169], will therefore be of paramount importance in establishing the feasibility and scope of AI implementation in clinical practice [154].

## 9. Conclusions

Deep learning stands at the forefront of advancements in breast cancer imaging, offering unparalleled potential to improve diagnostic accuracy and provide new opportunities in terms of prognosis and response prediction. Despite remarkable progress made in recent years, challenges remain on the path towards widespread clinical adoption.

The limitations surrounding generalizability and interpretability underscore the need for continued research in algorithm development and AI integration strategies. Efforts to address these challenges through the acquisition of diverse, high-quality datasets, the development of more easily interpretable and explainable models, the implementation of multimodal analysis techniques, and the enhancement of robustness and scalability will be imperative for maximizing the utility of AI in clinical practice.

Concerns about availability, explainability, costs, and privacy highlight the need for public datasets and open training and inference platforms for worldwide developers and healthcare professionals.

Collaborative research from hardware and software vendors, clinicians, and policymakers will also be required to improve computational infrastructures, enhance data security measures, promote responsible use by radiologists, and confront complex bioethical implications.

Despite existing challenges, the role of AI in breast cancer imaging cannot be underestimated. With ongoing interdisciplinary collaboration, rigorous validation studies, and a commitment to addressing current limitations, deep learning holds the potential to significantly improve diagnostic accuracy, personalize treatment strategies, and ultimately enhance patient outcomes in the fight against breast cancer.

## Figures and Tables

**Figure 1 diagnostics-14-00848-f001:**
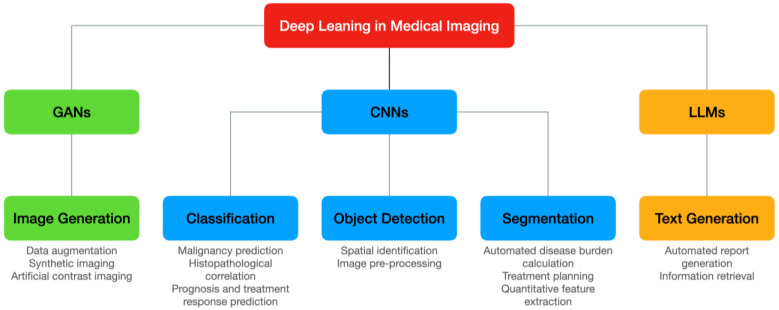
Schematic representation of deep learning applications in medical imaging. GANs: generative adversarial networks. CNNs: convolutional neural networks. LLMs: large language models.

**Figure 2 diagnostics-14-00848-f002:**
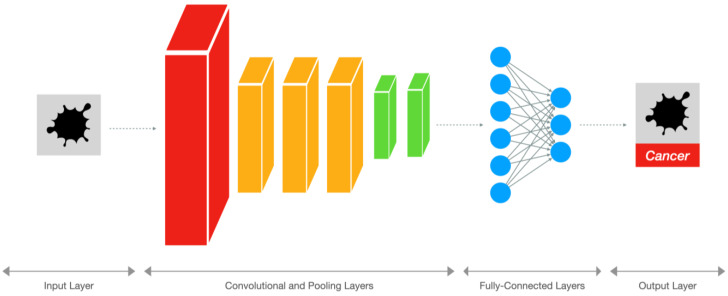
Simplified illustration of a convolutional neural network (CNN) architecture for image classification. Blocks of convolutional layers extract features from the image by applying weighted filters able to detect edges, shapes, textures, and patterns. Fully connected layers perform the final prediction, labeling the image under a specific category.

**Figure 3 diagnostics-14-00848-f003:**
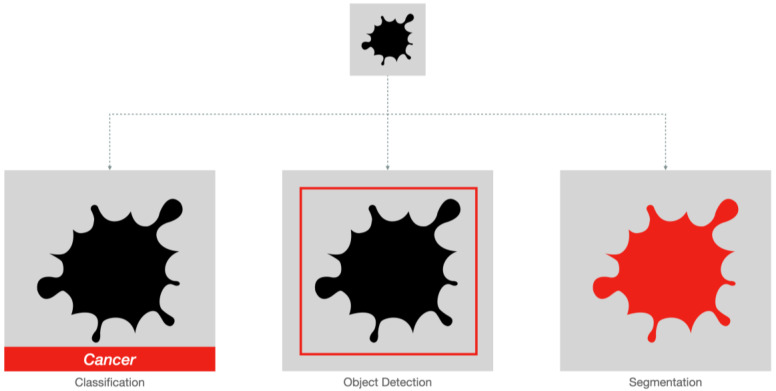
Inference output for medical imaging-related computer vision tasks. Classification labels the image under a specific category. Object detection draws bounding boxes containing abnormalities. Segmentation identifies the exact area or volume occupied by abnormalities.

**Figure 4 diagnostics-14-00848-f004:**
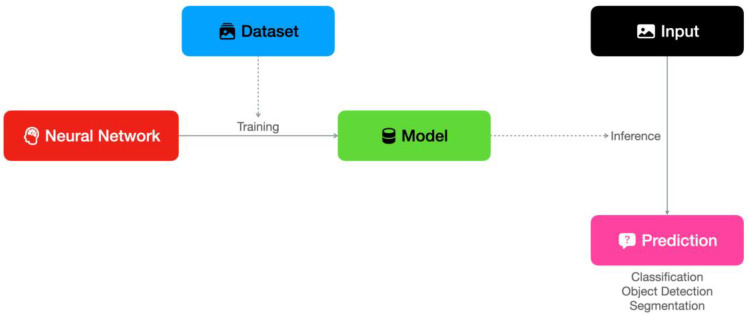
Simplified illustration of a typical deep learning-based model development workflow. A neural network is trained using annotated, ground-truth information from a dataset, generating a trained model. The trained model can then be used for inferring predictions on new input data.

**Figure 5 diagnostics-14-00848-f005:**
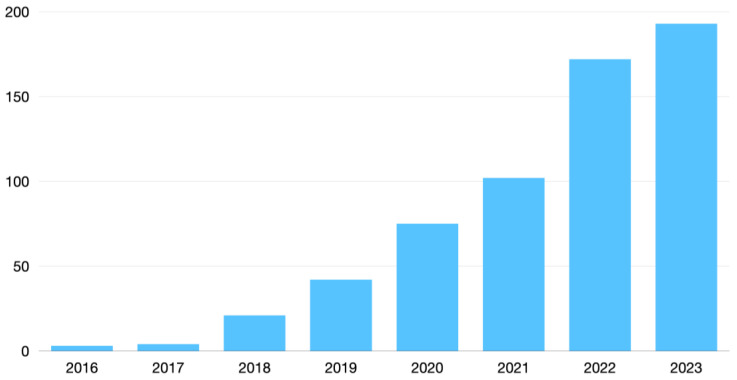
Number of articles indexed in PubMed^®^ by year with title/abstract containing *deep learning*, *breast*, *cancer*, and *imaging* as keywords.

**Figure 6 diagnostics-14-00848-f006:**
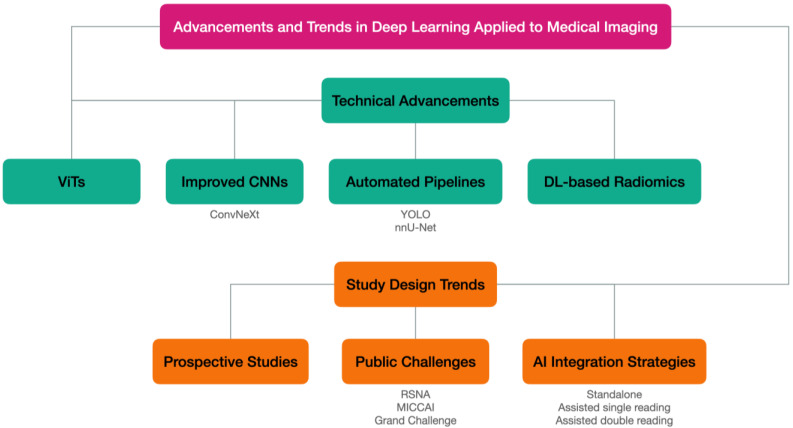
Summary of recent technical and study design advancements in medical imaging-applied deep learning research. ViTs: vision transformers. CNNs: convolutional neural networks. YOLO: You-Only-Look-Once. nnU-Net: No-New-U-Net. DL: deep learning. AI: artificial intelligence. RSNA: Radiological Society of North America. MICCAI: Medical Image Computing and Computer Assisted Intervention.

**Figure 7 diagnostics-14-00848-f007:**
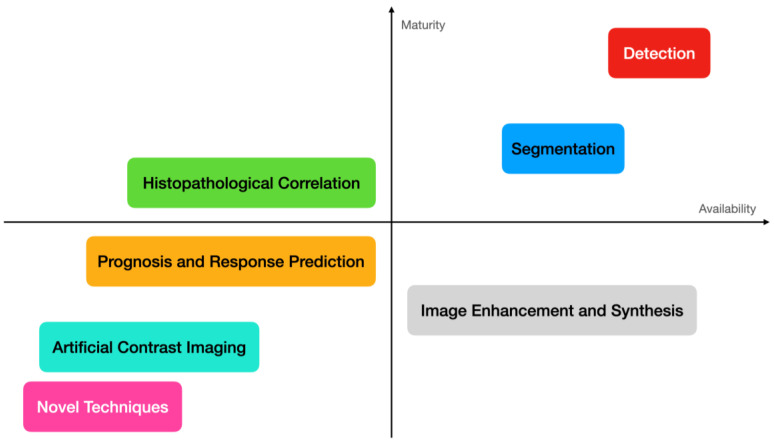
Status of deep learning applications for breast cancer imaging in early 2024. Inspired by Taylor et al. (2023) [9].

**Figure 8 diagnostics-14-00848-f008:**
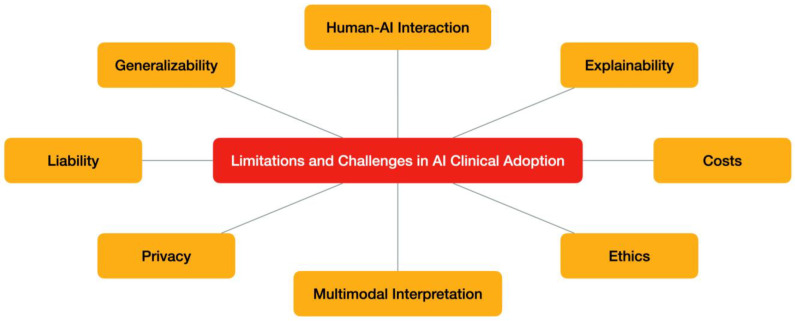
Summary of principal limitations in clinical adoption of AI-based software platforms. AI: artificial intelligence.

**Table 1 diagnostics-14-00848-t001:** Principal public datasets for breast cancer imaging examinations. SFM: screen-film mammography. FFDM: full-field digital mammography. US: ultrasound. DCE-MRI: dynamic contrast-enhanced magnetic resonance imaging. DDSM: Digital Database for Screening Mammography. CBIS-DDSM: curated breast imaging subset of DDSM. ADMANI: Annotated Digital Mammograms and Associated Non-Image.

Dataset	Origin	Release Year	Number of Patients	Modality
**DDSM**	United States	1999	2620	SFM
**INBreast**	Portugal	2011	115	FFDM
**CBIS-DDSM**	United States	2017	1566	SFM (improved)
**VinDr-Mammo**	Vietnam	2022	5000	FFDM
**ADMANI**	Australia	2022	630,000 (40,000 public test images)	FFDM
**BrEaST**	Poland	2023	256	US
**BUS-BRA**	Brazil	2023	1064	US
**Duke-Breast-Cancer-MRI**	United States	2022	922	DCE-MRI
**BreastDM**	China	2023	232	DCE-MRI

**Table 2 diagnostics-14-00848-t002:** Key studies involving deep learning and conventional breast cancer imaging techniques. dANN: deep artificial neural network. ML: machine learning. DL: deep learning. DLR: deep learning-based radiomics. MG: mammography. DBT: digital breast tomosynthesis. CEM: contrast-enhanced mammography. MRI: magnetic resonance imaging. DCE-MRI: dynamic contrast-enhanced MRI.

Authors	Year	Software/Model	Modality	Type	Task
**Becker et al. [76]**	2017	dANN (ViDi 2.0)	MG	Retrospective	Classification
**Watanabe et al. [77]**	2019	cmAssist^®^	MG	Retrospective	Classification
**Akselrod-Ballin et al. [78]**	2019	Custom ML + DL	MG	Retrospective	Classification
**Schaftter et al. [79]**	2020	Multiple (public challenge)	MG	Retrospective	Classification
**Kim et al. [80]**	2020	INSIGHT MMG	MG	Retrospective	Classification
**Dembrower et al. [81]**	2020	INSIGHT MMG	MG	Retrospective	Classification
**Dembrower et al. [82]**	2023	INSIGHT MMG 1.1.6	MG	Prospective	Classification
**Ng et al. [83]**	2023	Mia^®^ 2.0 (Kheiron Medical Technologies Ltd., London, UK)	MG	Prospective	Classification
**Romero-Martín et al. [84]**	2021	Transpara^®^	MG, DBT	Retrospective	Classification
**Zheng et al. [85]**	2023	RefineNet + Xception	CEM	Prospective	Segmentation, Classification
**Beuque et al. [86]**	2023	DL + handcrafted radiomics	CEM	Retrospective	Segmentation, Classification
**Qian et al. [87]**	2023	Multi-feature fusion network	CEM	Retrospective	Classification
**Gu et al. [88]**	2022	VGG19	US	Prospective	Classification
**Janse et al. [89]**	2023	nnU-Net	DCE-MRI	Retrospective	Segmentation
**Li et al. [90]**	2023	Custom DLR	DCE-MRI	Retrospective	Treatment Response Prediction

**Table 3 diagnostics-14-00848-t003:** Principal FDA-approved AI-based tools for breast cancer detection. MG: mammography. DBT: digital breast tomosynthesis.

Product	Vendor	Country	Modality
**cmAssist^®^**	CureMetrix Inc., La Jolla, CA, USA	United States	MG
**Genius AI Detection**	Hologic Inc., Marlborough, MA, USA	United States	MG and DBT
**INSIGHT MMG**	Lunit Inc., Seoul, Republic of Korea	South Korea	MG
**MammoScreen^®^ 2.0**	Therapixel SA, Nice, France	France	MG and DBT
**ProFound AI^®^**	iCAD Inc., Nashua, NH, USA	United States	MG and DBT
**Saige-Dx**	DeepHealth Inc., Somerville, MA, USA	United States	MG
**Transpara^®^**	ScreenPoint Medical B.V., Nijmegen, The Netherlands	Netherlands	MG and DBT

**Table 4 diagnostics-14-00848-t004:** Summary of designs, results and limitations of the screening mammogram cancer detection studies included in this review. AUC: area under the receiver operating characteristic curve.

Authors	Study Design	Key Results	Highlighted Limitations
**Becker et al. [76]**	Standalone classifier for breast cancer detection versus experienced radiologists	AUC = 0.81 on first training dataset, 0.79 on external testing cohort, 0.82 on second, screening-like cohort (statistically equivalent to experienced radiologists)	Not true screening cohort and retrospective design leading to potential selection bias; worse specificity than experienced radiologists; no understanding of laterality and time evolution, and no inclusion of clinical and bioptic data in the algorithm
**Watanabe et al. [77]**	Radiologist-paired classifier for breast cancer detection to improve radiologists’ sensitivity	Overall reader CDR increased from mean of 51% to mean of 62% (mean of 27% relative increase)	Cancer-enriched dataset and retrospective design leading to potential selection bias; lack of comparison of prior mammograms for radiologists, funding by AI software company
**Akselrod-Ballin et al. [78]**	Standalone classifier for breast cancer detection (malignancy prediction)	AUC = 0.91 with specificity of 77.3% at a sensitivity of 87%	Selection bias; single mammography scanner vendor potentially limiting generalizability; many patients excluded after a single negative examination; distinction between screening and diagnostic studies not well defined; no lesion localization
**Schaftter et al. [79]**	Standalone and radiologist-paired classifier for breast cancer detection; public challenge	Standalone: AUC = 0.858 and 0.903 on the internal and external validation dataset, respectively; radiologist-paired: AUC = 0.942	Interaction between radiologists and AI not well studied; larger training and validation datasets not available for challenge participants; no cancer spatial annotation; small number of positive cases
**Kim et al. [80]**	Standalone classifier for breast cancer detection versus unassisted and AI-assisted radiologists	AUC = 0.940 for standalone AI versus 0.810 for unassisted radiologists and 0.881 for assisted radiologists, with better performance in detection of mass, distortion, asymmetry, and T1 and node-negative cancers	Cancer-enriched dataset and retrospective design; clinical factors not considered by the algorithm; reading setting potentially different from clinical practice; funding by AI software company
**Dembrower et al. [81]**	Standalone classifier for screening mammograms triage	Missed cancers: 0, 0.3%, or 2.6% for 60%, 70%, or 80%-lowest AI score rule-out, respectively; additional interval cancer detection: 12% or 27% for 1% or 5%-high AI score rule-in, respectively; additional next-round cancer detection: 14% or 35%, respectively	Retrospective design; screening cohort not fully examined; previous mammogram within 30 months before diagnosis required for inclusion; no cancer spatial annotation; single demographic; biennial screening program; interaction between radiologists and AI not well studied; arbitrary triage cut-offs
**Dembrower et al. (2023) [82]**	Radiologist-paired classifier (assisted single reading) for breast cancer detection versus standalone classifier, unassisted double reading, and assisted double reading (*triple reading*); prospective, non-inferiority study	Non-inferiority of both assisted single reading and standalone classifier compared to double reading	Availability of both AI and radiologists results in the consensus discussion, potentially underestimating AI ability; abnormality threshold based on retrospective data; no quality assurance mechanisms implemented; single-arm paired design preventing comparison of differences in interval cancer rates; no biopsy for negative screening examinations; single scanning machine vendor and AI software used potentially limiting generalizability, funding by AI software company
**Ng et al. [83]**	Radiologist-paired classifier (*assisted double reading*) for breast cancer detection versus unassisted double reading; prospective study	Additional 0.7–1.6 cancer detection per 1000 cases, with 0.16–0.30% additional recalls, 0–0.23% unnecessary recalls, and 0.1–1.9% increase in positive predictive value; majority of extra detected cancers featuring invasiveness and small size	Data collected from a single breast cancer institution; only one commercial AI software evaluated; short follow-up (2 to 9 months) preventing evaluation of interval cancer rates; unclear impact of inter-reader variation when introducing AI in the process, funding by AI software company

**Table 5 diagnostics-14-00848-t005:** Key potential advantages and current challenges of novel breast imaging techniques.

Potential	Challenges
Non-ionizing radiation	Limited availability of scan devices
No breast compression	Lack of standardized protocols
Lower costs	Limited availability of curated datasets
Lower performance loss in dense breasts	Lack of studies in clinical and screening settings

**Table 6 diagnostics-14-00848-t006:** Key studies involving deep learning and novel breast cancer imaging techniques. CNN: convolutional neural network. FNN: feed-forward neural network. DNN: deep neural network. ML: machine learning. DL: deep learning. TG: thermography. USE: ultrasound elastography. AE: autoencoder. DOT: diffuse optical tomography. US: ultrasound.

Authors	Year	Software/Model	Modality	Type	Task
**Mambou et al. [108]**	2017	Custom ML + DL	TG	Retrospective	Classification
**Mohammed et al. [109]**	2021	InceptionV4	TG	Retrospective	Classification
**Alshehri et al. [110]**	2022	Custom CNN + AM	TG	Retrospective	Classification
**Mohamed et al. [112]**	2022	U-Net + bespoke classifier and VGG16	TG	Retrospective	Segmentation, Classification
**Civiliban et al. [113]**	2023	Mask R-CNN	TG	Retrospective	Segmentation
**Khomsi et al. [114]**	2024	Custom FNN	TG	Retrospective	Tumor size estimation
**Singh et al. [115]**	2021	Thermalytix^®^	TG	Prospective	Classification
**Bansal et al. [116]**	2023	Thermalytix^®^	TG	Prospective	Classification
**Zhang et al. [126]**	2015	DNN feature extractor + ML classifier	USE	Retrospective	Classification
**Fukuda et al. [127]**	2023	GoogLeNet	USE	Retrospective	Classification
**Yu et al. [130]**	2023	ResNet18	BSGI	Retrospective	Classification
**Zhang et al. [135]**	2023	Custom fusion model (VGG11 + AE)	DOT + US	Retrospective	Classification

## Data Availability

No new data were created or analyzed in this study. Data-sharing rules are not applicable to this article.

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
