# Peer review of "Deep Learning in Breast Cancer Imaging: State of the Art and Recent Advancements in Early 2024"

_diagnostics, 2024, doi:10.3390/diagnostics14080848_

Round 1
Reviewer 1 Report
Comments and Suggestions for Authors
In this paper, authors presented a review of breast cancer imaging using deep learning based approaches. However, the paper is lacking in certain key aspects. Following are my observations:
1. Table 2 is not cited anywhere in the text.
2. The plag report shows quite a chunk of text which matches exactly with the existing studies. Authors are advised to take this into serious consideration and rectify this.
3. Section 2: Deep Learning in medical Imaging is a generalized section with respect to the topic of the current review in hand, as it covers the existing literature for use of deep learning in various roles for medical imaging domain. There alot of surveys already existing which elaborates the same. So, authors might want to cut short this section and instead give a foundational knowledge or introduction on Breast Cancer Imaging, different modalities used for this, in what roles deep learning has been used in breast cancer imaging, etc.
4. Kindly, illustrate the research gaps and challenges in bullets or numbering format to clearly indicate the existing gaps and challenges present in the area of breast cancer imaging.
5. There is no mention of current accuracy or any empirical results achieved by existing DL methods for breast cancer imaging. Without this, how authors can conclude the future scope or challenges exiting in the area of breast cancer detection.
6. There is no comparison of the presented review with the existing surveys on breast cancer imaging. Kindly, illustrate how the information in the presented review is different form those already available in the existing reviews. Authors need to differentiate how their review is different from these. Authors need to show point wise comparison of their review with these to substantiate the need of their work.
7. The author needs to provide sumarized tables for every sub section mentioned in their review, clearly illustrating the objective of the work, what type of DL methods have been used for breast cancer, key results (if any) and limitations or remarks. These would enable the viewers to have clear picture of the trend being followed for breast cancer imaging and the highly adopted DL methods for the same in various sub-areas of breast cancer imaging.
Author Response
Dear reviewer,
Thank you for the time you dedicated to reading our manuscript and the valuable feedback you have provided us with.
First, we would like to clarify the goal and scope of our review. We have written this manuscript to provide radiologists with an updated overview of AI techniques for breast cancer imaging, along with their potential, limitations and current challenges. We noted a lack of fundamental technical knowledge about AI implementation among radiologists, and we feel this needs to be corrected to give them a better understanding of both final applications and related literature. Therefore, we included a dedicated section (Section 3) that tries to address this issue using a reasonably simple technical discussion with accompanying figures. Moreover, considering the fast-paced evolution of this field, we wanted to report up-to-date information about recent developments and trends, both on the technical and clinical research side. We acknowledge many articles on this topic have already been published, but we think our review provides value in exploring these two fundamental aspects.
We hope this clarification will help you better understand the organization of our review article.
Here is a summary of the main changes included in the latest revision:
- Tables and figures have been tweaked and are now appropriately cited in the text
- We checked the body for repeated abbreviations and expansions
- We improved Section 1 (Introduction) by better detailing the role of AI in breast cancer imaging and briefly citing its evolution
- The longest text sections have been slightly simplified while retaining essential information
- Section 8 (Discussion) has been split in two sub-sections for state-of-the-art and current challenges, and the latter has been enhanced providing a future direction for each limitation and organized in separated points for better readability
This is the point-by-point response to the comments and observations:
1) Table 2 is now cited at line 393.
2) We have slightly simplified the report of previous clinical studies in Section 5 and 6.
3) We have slightly simplified Section 3 (formerly 2). However, our main goal was to give an updated introduction of breast imaging AI techniques to radiologists. We privileged discussing fundamental technical concepts in AI for medical imaging, since we feel these are lesser known among our target audience despite being necessary to understand both AI instruments and related literature. We have stated this purpose in Section 2. Conversely, we avoided discussing technical details about imaging modalities since our audience should already be familiar with them.
4) We have now organized key limitations and challenges by bullet points in Section 8.2; at the end of each of them we provided a relevant future research direction. Figure 8 also provides a schematic overview of current issues.
5) We have reworked Section 8, splitting it in a brief review of current state of the art and empirical results achieved in different applications (Section 8.1) and a discussion of limitations, challenges and future directions (Section 8.2).
6) We have now included our Contribution, Novelty, and Motivation Statement in Section 2.
7) We have reworked the included tables and figures:
- Figure 1, 2, 3, and 4 cover the technical introduction (Section 3)
- Table 1 covers key breast cancer imaging datasets (Section 4)
- Figure 5, Table 2 and Table 3 cover DL applied to conventional breast cancer imaging (Section 5)
- Table 5 and 6 cover DL applied to novel breast imaging techniques (Section 6)
- Figure 6 covers recent advancements in DL applied to medical and breast cancer imaging (Section 7).
- Figure 7 and 8 cover the final discussion (Section 8).
Reviewer 2 Report
Comments and Suggestions for Authors
This study aims to review the role of deep learning (DL) in breast cancer imaging, with the intention of stimulating clinical interest and technological advancements for diagnostics.
Firstly, the overall length of this review is extensive, but the specific focus remains unclear. In some places, it appears the authors intend to outline a workflow for readers on implementing DL tools in their own studies, while in other places it appears as they aim to retrospect the development and progression of DL models in the history of breast cancer imaging analysis. Even within the latter scope, it remains ambiguous whether the authors adopt a clinical-based perspective or a technical (algorithm)-based viewpoint. I think the whole manuscript needs to be redrafted and I recommend the authors to narrow the scope to either of the aforementioned focuses.
Secondly, reading this article is more akin to perusing a dictionary, which simply offers: (i) a list of numerous words related to AI and medical imaging with explanations, and (ii) a compilation of recently published papers related to AI and breast cancer imaging. The primary goal of a review article should be to provide researchers with an overarching understanding of the background leading to their research questions, pertinent findings that may inform their inquiries, and unresolved issues for future investigation. However, throughout the manuscript, the authors fail to offer these insights.
Additional comments:
1. The introduction section is too general. To align with the title, the authors should at least provide a concise overview of the development and progression of AI in breast cancer imaging, as well as outline the advantages of AI over traditional methods.
2. The figures are simplistic, and some lack accuracy. For example, the depiction of convolutional layers in Figure 1 is inaccurate, and the workflow presented in Figure 4 is confusing. It is essential to include accurate icons and elements with detailed explanations in the captions to enhance clarity.
3. In some tables, such as Table 2 and 3, the nationality of the authors and the published year may not be of significant interest to most readers. Rather, readers are more likely to focus on aspects such as the DL methodologies employed, the performance of the study, and its associated pros and cons. Unfortunately, the authors did not provide this relevant information.
4. The “challenge” section is vital in a review article. I recommend that the authors divide this section into several sub-sections to systematically depict the current limitations point-by-point and anticipate potential solutions to these challenges.

Author Response
Dear reviewer,
Thank you for the time you dedicated to reading our manuscript and the valuable feedback you have provided us with.
First, we would like to clarify the goal and scope of our review. We have written this manuscript to provide radiologists with an updated overview of AI techniques for breast cancer imaging, along with their potential, limitations and current challenges. We noted a lack of fundamental technical knowledge about AI implementation among radiologists, and we feel this needs to be corrected to give them a better understanding of both final applications and related literature. Therefore, we included a dedicated section (Section 3) that tries to address this issue using a reasonably simple technical discussion with accompanying figures. Moreover, considering the fast-paced evolution of this field, we wanted to report up-to-date information about recent developments and trends, both on the technical and clinical research side. We acknowledge many articles on this topic have already been published, but we think our review provides value in exploring these two fundamental aspects.
We hope this clarification will help you better understand the organization of our review article.
Here is a summary of the main changes included in the latest revision:
- Tables and figures have been tweaked and are now appropriately cited in the text
- We checked the body for repeated abbreviations and expansions
- We improved Section 1 (Introduction) by better detailing the role of AI in breast cancer imaging and briefly citing its evolution
- The longest text sections have been slightly simplified while retaining essential information
- Section 8 (Discussion) has been split in two sub-sections for state-of-the-art and current challenges, and the latter has been enhanced providing a future direction for each limitation and organized in separated points for better readability
This is the point-by-point response to the comments and observations:
As we have stated in Section 2, this dual point of view was intentional. From our personal experience, we believe our target audience (radiologists) would benefit from a technical introduction as a preparatory step for understanding AI-related clinical research.
We have improved Section 1 (Introduction) to include a clearer background of AI role and evolution in medical imaging, and Section 8 (Discussion) to better represent state-of-the-art and existing challenges.
1) We have edited the Introduction including additional information about the role of AI in breast cancer imaging and briefly citing its evolution.
2) We tweaked the CNN illustration (Figure 2) mentioning pooling layers, and we added icons in Figure 4 to enhance clarity. We acknowledge our figures may appear excessively simplified to seasoned AI engineers, but we wanted to avoid excessive details that our primary target audience would find hard to understand.
3) We have tweaked Table 2 by including the software/AI model used in each study. Further details regarding key studies are mentioned in Table 4 and in the main body. Regarding Table 3, we decided to leave the nationality field of the software vendor because of its potential implications on performance across different patient demographics. Moreover, since the referenced commercial solutions are proprietary and closed source, we weren’t able to find detailed information about their internal architecture.
4) We have split Limitations, Challenges and Future Directions (Section 8.2) in individual subsection exploring each of the main issues and providing future research indications.
Reviewer 3 Report
Comments and Suggestions for Authors
Researchers examined the deep learning models in the literature for breast cancer detection from images and discussed current developments. The study is comprehensive enough, its results and challenges are presented with accurate analysis and discussed in detail. However, eliminating the issues listed below will make the study better quality.
1. In the Introduction, researchers mentioned the traditional methods used for breast cancer detection. What are the disadvantages of the methods? Why are artificial intelligence supported systems needed? Information about this should be given. What is the difference between artificial intelligence and these methods? Why is it more effective or why is it used effectively? These must be addressed.
2. None of the figures and tables in the article are referenced in the article. A brief explanation should be given before the figures and tables and the relevant figure or table should be stated.
3. Attention should be paid to abbreviations. For example, while the expression deep learning (DL) was given in the abstract, a similar situation was observed in line 83. The meaning of the DL expression has already been given before. There is no need to give the expansion again here. Similar situations exist elsewhere. The article should be read in order to eliminate these errors.
4. In line 108, the researchers mentioned max-pooling and avg-pooling layers. In addition to these, the min-pooling layer is also used.
5. Area Under Curve expression should be changed to (AUC). The expression AUC-ROC is not a correct usage. If there is such a use, I would appreciate if you could send me the relevant publication.
6. The conclusion of such a detailed review article is very inadequate. The conclusion should be enhanced and not only the impact of artificial intelligence, but also its ethical status and reliability in the field of health should be evaluated.
Comments on the Quality of English LanguageResearchers examined the deep learning models in the literature for breast cancer detection from images and discussed current developments. The study is comprehensive enough, its results and challenges are presented with accurate analysis and discussed in detail. However, eliminating the issues listed below will make the study better quality.
1. In the Introduction, researchers mentioned the traditional methods used for breast cancer detection. What are the disadvantages of the methods? Why are artificial intelligence supported systems needed? Information about this should be given. What is the difference between artificial intelligence and these methods? Why is it more effective or why is it used effectively? These must be addressed.
2. None of the figures and tables in the article are referenced in the article. A brief explanation should be given before the figures and tables and the relevant figure or table should be stated.
3. Attention should be paid to abbreviations. For example, while the expression deep learning (DL) was given in the abstract, a similar situation was observed in line 83. The meaning of the DL expression has already been given before. There is no need to give the expansion again here. Similar situations exist elsewhere. The article should be read in order to eliminate these errors.
4. In line 108, the researchers mentioned max-pooling and avg-pooling layers. In addition to these, the min-pooling layer is also used.
5. Area Under Curve expression should be changed to (AUC). The expression AUC-ROC is not a correct usage. If there is such a use, I would appreciate if you could send me the relevant publication.
6. The conclusion of such a detailed review article is very inadequate. The conclusion should be enhanced and not only the impact of artificial intelligence, but also its ethical status and reliability in the field of health should be evaluated.
Author Response
Dear reviewer,
Thank you for the time you dedicated to reading our manuscript and the valuable feedback you have provided us with.
First, we would like to clarify the goal and scope of our review. We have written this manuscript to provide radiologists with an updated overview of AI techniques for breast cancer imaging, along with their potential, limitations and current challenges. We noted a lack of fundamental technical knowledge about AI implementation among radiologists, and we feel this needs to be corrected to give them a better understanding of both final applications and related literature. Therefore, we included a dedicated section (Section 3) that tries to address this issue using a reasonably simple technical discussion with accompanying figures. Moreover, considering the fast-paced evolution of this field, we wanted to report up-to-date information about recent developments and trends, both on the technical and clinical research side. We acknowledge many articles on this topic have already been published, but we think our review provides value in exploring these two fundamental aspects.
We hope this clarification will help you better understand the organization of our review article.
Here is a summary of the main changes included in the latest revision:
- Tables and figures have been tweaked and are now appropriately cited in the text
- We checked the body for repeated abbreviations and expansions
- We improved Section 1 (Introduction) by better detailing the role of AI in breast cancer imaging and briefly citing its evolution
- The longest text sections have been slightly simplified while retaining essential information
- Section 8 (Discussion) has been split in two sub-sections for state-of-the-art and current challenges, and the latter has been enhanced providing a future direction for each limitation and organized in separated points for better readability
This is the point-by-point response to the comments and observations:
1) We improved the introduction (Section 1) by better detailing the role of AI in breast cancer imaging.
2) All figures and tables are now appropriately referenced in the text.
3) We have checked the text for repeated expansions. We hope to have addressed all of them.
4) We have now cited the min-pooling approach for pooling layers.
5) We have replaced occurrences of AUC-ROC with AUC.
6) We have enhanced Discussion and Conclusion (Section 8 and 9) providing a comment and future direction for each of the key limitations we identified. Emerging themes such as privacy, human-AI interaction, explainability, ethics and liability have also been cited.
Round 2
Reviewer 2 Report
Comments and Suggestions for Authors
The authors have addressed my concerns.
Reviewer 3 Report
Comments and Suggestions for Authors
The authors made the requested revisions and enhanced the article. The article can be accepted in this form.